# Wheat Genes Associated with Different Types of Resistance against Stem Rust (*Puccinia graminis* Pers.)

**DOI:** 10.3390/pathogens11101157

**Published:** 2022-10-07

**Authors:** Anatolii Karelov, Natalia Kozub, Oksana Sozinova, Yaroslav Pirko, Igor Sozinov, Alla Yemets, Yaroslav Blume

**Affiliations:** 1Institute of Food Biotechnology and Genomics, National Academy of Sciences of Ukraine, 04123 Kyiv, Ukraine; 2Institute of Plant Protection, National Academy of Agrarian Sciences of Ukraine, 03022 Kyiv, Ukraine

**Keywords:** stem rust, *Puccinia graminis* Pers., Ug99, wheat, resistance genes, adult plant resistance

## Abstract

Stem rust is one wheat’s most dangerous fungal diseases. Yield losses caused by stem rust have been significant enough to cause famine in the past. Some races of stem rust are considered to be a threat to food security even nowadays. Resistance genes are considered to be the most rational environment-friendly and widely used way to control the spread of stem rust and prevent yield losses. More than 60 genes conferring resistance against stem rust have been discovered so far (so-called *Sr* genes). The majority of the *Sr* genes discovered have lost their effectiveness due to the emergence of new races of stem rust. There are some known resistance genes that have been used for over 50 years and are still effective against most known races of stem rust. The goal of this article is to outline the different types of resistance against stem rust as well as the effective and noneffective genes, conferring each type of resistance with a brief overview of their origin and usage.

## 1. Introduction

Biotrophic phytopathogenic fungi are obligate parasites of plants that during evolution developed the ability to penetrate host cells without destruction for obtaining nutrients and energy [1]. Rust fungi of bread wheat (*Triticum aestivum* L.) cause diseases such as leaf rust (caused by *Puccinia recondita* Dietel and Holw), yellow rust (*Puccinia striiformis* var. *striiformis* Westend), and stem rust (*Puccinia graminis* Pers.), which may seriously affect wheat yield worldwide [2]. For instance, yield losses could be up to 100% for especially pathogenic races of stem rust [3]. Significant yield losses related to epiphytotics of stem rust were reported in Australia, the USA, Scandinavian countries, Central and South Europe, India, and Asia in the 20th century [4,5]. The situation became even more dramatic in the 21st century starting with stem rust epidemics in Africa. Furthermore, in the last decade there have been significant outbreaks of stem rust in Kenya due to the emergence of new Ug99 races [6], epidemics and devastating yield losses in Ethiopia in 2013 due to the TKTTF race [3], outbreaks in North Kazakhstan and Siberia in 2015–2017 [7,8], epidemics of stem rust in Germany in 2013 [9], the first cases of wheat stem rust infection in the United Kingdom in nearly 60 years [10], an outbreak of stem rust in Southern Italy in durum wheat [11], and further spread of stem rust in Europe [12].

Losses caused by the disease may be explained by the details of the life cycle and pathogenesis of *P. graminis*. The life cycle of the fungus involves five different spore stages during the asexual reproduction in wheat (the uredinial stage) and sexual reproduction, which starts at the teliospore stage and continues on an alternate host plant (barberry, Mahonia). Ascospores complete the *P. graminis* life cycle infecting cereals [13]. This process is associated with the formation of urediniospores positioned on the surface of a leaf sheath or a stem and further development of the complex system of penetration of the plant cell, which includes appresoria, a penetration peg, hyphae, haustoria, and a substomatal vesicle to provide nutrients to the parasite. In areas with mild winters and sufficiently wet springs, *P. graminis* can exist in the uredinial (asexual) state in winter cultivated and wild cereals [4,13]. In the case of significant stem rust infestation of plants, nutrient flow to kernels can be affected causing shriveled grain. Moreover, stems are weakened by the disease resulting in wheat lodging, which causes additional yield losses [13].

Another factor that makes stem rust an especially dangerous disease for wheat is its polymorphism and ability for mutagenesis of the causative agent and the rapid emergence of new *P. graminis* races, as it is a species with a high-evolutionary potential [14]. Regularly, shortly after the wide implementation of a gene conferring resistance to the disease, a race virulent to that gene emerges causing significant losses to agriculture in some countries [15,16]. Only a few stem rust genes have shown durable effectiveness in breeding history. One of these genes is the stem rust resistance gene on translocation 1BL/1RS from the Petkus rye (*Secale cereale* L.), which reliably provided stem rust resistance for about 40 years until the emergence of the first race of the Ug99 group, TTKSK, with virulence to *Sr31* in Uganda in 1999, which turned out to be virulent to the majority of other widespread resistance genes [15]. Despite preventive measures to localize Ug99, it spread to the southern coast of Africa. In addition, the original race TTKSK has been reported in the Middle East [16]. Moreover, new types (probably mutants) of Ug99 have been detected, which have gained the status of races. In particular, especially virulent not only to *Sr31* but also to other genes that, according to initial studies, conferred resistance to Ug99, are the races TTKST, TTTSK, TTKSP, PTKSK, PTKST, TTKSF+, TTKTT, TTKTK, TTHSK, TTHST, PTKTK, TTKTT+, and TTHTT discovered from 2005 to 2020 in Tanzania, Eritrea, Egypt, Rwanda, Kenya, Ethiopia, South Africa, Yemen, Mozambique, Zimbabwe, and Uganda [6,15,16,17,18,19,20,21]. In 2019, the race TTKTT was reported in Iraq [22]. The last decade is characterized by stem rust outbreaks in Europe, Asia, and African regions due to the emergence of new stem rust races with multiple virulences that are distinct from the Ug99 group [3,7,8,9,12,23,24,25,26]. The Digalu race (TKTTF) caused severe epidemics in southern Ethiopia in 2013–2014 when yield losses were up to 100% of the wheat cultivar ‘Digalu’ planted in large areas [3]. Among the currently prevalent European races are TTRTF, TKTTF, and TKKTF. The race TTRTF caused the outbreak of stem rust in Sicily in 2016 [27]. This race was first described in 2014 in Georgia [23] and became widespread in Europe [12]. TTRTF was also detected in 2016 in Eritrea [25] and 2019 in Ethiopia [24] and the south of Iran [25]. This race is avirulent to *Sr31* but has virulence to many important genes providing resistance against Ug99 races such as *Sr13b, Sr35, Sr37,* and *Sr50* [12,23]. Moreover, a number of novel stem rust races with virulence to *Sr31* and other stem rust genes have been recently described including TKHBK [26] and 22 other races in Spain [12] and the race LTBSK in Western Siberia [12].

Genes conferring resistance against stem rust are referred to as *Sr* genes [5]. More than 60 genes have been identified to date. Some of them were detected in bread wheat (subsequently referred to as wheat own genes) and others were introgressed from related species. The majority of *Sr* genes are seedling (all-stage or juvenile) resistance genes. A small number of genes belong to race-nonspecific adult plants resistance genes (APRs). The goal of this article is to outline the different types of resistance against stem rust as well as the effective and noneffective genes, conferring each type of resistance with a brief overview of their origin and usage.

## 2. Own Resistance Genes in Bread Wheat

The majority of widespread own stem rust resistance genes of wheat are neither effective against races of stem rust that are currently common throughout the world, nor do they confer resistance against the especially dangerous races of the Ug99 group (Table 1) [5,16,17,18,19,20]. For instance, the resistance gene *Sr5* on chromosome 6DS originated from the cultivar ‘Kanred’ and developed on the basis of the Ukrainian (Crimean) gene pool, is quite common among modern wheat cultivars [5,28]. Initially the gene conferred race-specific immunity-like resistance. However, cultivars with this gene had been cultivated in large areas so subsequently a number of *P. graminis* races were able to overcome *Sr5* [29,30].

Another wheat own stem rust resistance gene, *Sr6* on chromosome 2D, is also quite common. The gene was identified in the Canadian cultivar ‘McMurachy’ and most likely derives from the African wheat gene pool [5,21]. The level of resistance conferred by *Sr6* depends on the environmental conditions [31]. Currently many stem rust races are virulent to the gene [29,30]. The *Sr7* gene (with alleles *a* and *b*) is located on chromosome 4AL [32]. The allele *a* of the gene was first found in some cultivars from Kenya [28,32]. The resistance level conferred by *Sr7* is also largely dependent on environmental conditions and genetic background [33,34] and there are stem rust races with virulence to the allele *a* of this gene [29,30]. The allele *b* of *Sr7* was introduced into breeding from Australian wheat cultivars unintentionally in the 1920s and also originates from African bread wheat cultivars; the allele confers resistance to the stem rust races that are dominant in Australia [71] but not to Ug99 races [15], TTRTF [11], TKTTF, TKKTF, TKPTF, PKPTF, TKKTP [9] and some other races found in Europe [8,9,12] and Western Siberia [8]. The resistance conferred by the gene *Sr8* on chromosome 6AS is associated with the alleles *a* and *b* [35,36,37]. The allele *a* is widely represented among modern cultivars while the allele *b* is rarely encountered [5,28]. Both alleles confer a moderate level of stem rust resistance (in case of the allele *b*, the resistance is temperature-dependent), which is overcome by stem rust races that are common worldwide (including by some races that had been reported to be avirulent to it) [29].

The *Sr9* gene was localized on chromosome 2BL of wheat [49]. The alleles *a* and *b* of the gene originated from common wheat [39,49], but the allele *c* was transferred from *Triticum timopheevii* Zhuk. and further designated as *Sr36*, whereas the allele *d* was introgressed from *T. turgidum* subsp. *dicoccum* (Schrank) Schübl. [72] and *g* was from *T. turgidum* (L.) Thell. ssp. *durum* (Desf.) Husn. [5]. The bread wheat own allele *h* was initially designated as *SrWeb* as it derives from the Canadian cultivar Webster [41]. Moreover, it is one of few conferring resistance genes against most Ug99 races, except for TTKSF+ [21], but some other races of stem rust are virulent to this gene [41,72]. Other alleles of *Sr9* are more or less sensitive to widespread races of stem rust [5]. *Xwms47* is a molecular marker for the allele *h* of *Sr9* [41].

Some of the stem rust resistance genes of wheat are more effective under certain temperature conditions [40,44]. For instance, *Sr10*, a bread wheat own gene located on chromosome 2B, which was first found in the Kenyan gene pool of bread wheat, is quite common among cultivars developed in different regions in different periods of time [40,42]. The gene is effective under lower temperatures and was characterized as an APR gene [42] but it was not considered to be effective against the currently widespread *P. graminis* races [73].

The stem rust resistance gene *Sr15* was localized on chromosome 7AL, it is race-specific and not effective at temperatures higher than 26 °C [43,44]. *Sr15* cosegregates with the leaf rust resistance gene *Lr20* [45,46], the root lesion nematode resistance gene *Rlnn1*, and is closely linked to the powdery mildew resistance gene *Pm1* [46]. It was first identified in cv. ‘Norka’ but afterwards was found in cultivars that were not related to it [28,45]. There are many races with virulence to *Sr15* and the virulence level might be quite high [29,30]. Initially the gene was considered to confer no resistance against Ug99, but recent research has suggested otherwise [56]. The markers *wri1–5*, which were proposed to detect *Rlnn1*, might be also considered as diagnostic markers for *Sr15* [46].

The common wheat own gene *Sr16* was localized on chromosome 2BL [38,47]. The main source of *Sr16* is considered to be cv. ‘Reliance’, and it probably inherited the gene from cv. ‘Kanred’ [5,28]. There are not many modern races of stem rust that are avirulent to this gene [30]. The *Sr18* gene is also an ineffective own stem rust resistance gene; it is located on chromosome 1DL, and its origin is unknown [48,49]. The genes *Sr19* and *Sr20* originated from cv. ‘Marquiz’ and were localized on chromosome 2B [34]. None of them provide resistance against most races of *P. graminis* [29,73].

The *Sr23* gene is effective only at high temperatures and with sufficient lighting [51]. The gene is located on chromosome 2BS and cosegregates with the leaf rust resistance gene *Lr16* [51,52]. The sources of this gene are cv. ‘Selkirk’, ‘Exchange’, and ‘Warden’. The diagnostic markers for *Lr16* might also be used to detect the *Sr23* gene [28,52]. *Sr23* is effective against old races of stem rust from the Australian collection but not against modern races with few exceptions [29,73].

Some wheat own stem rust resistance genes were tested with races of the Ug99 group and showed different levels of effectiveness. The *Sr28* gene is located on chromosome 2BL and derives from cv ‘Kota’ [53]. Stem rust races that are virulent to this gene are quite common [29] and avirulent races mostly originate from Ethiopia and Nepal [30]. However, the result “moderate resistance–moderate sensitivity” was obtained while testing this gene against Ug99 in Njoro, Kenya in 2004–2005 [54]. In addition, according to literature, *Sr28* might confer moderate APR to the stem rust races BCCBC, TTKSK, and TTKST (the latter two belong to the Ug99 group) [55]. The markers *wPt-7004* and *wmc332* are considered to be diagnostic markers for this gene [56,57].

*Sr29* on chromosome 6D is a bread wheat own stem rust resistance gene of European origin [58,59]. The gene decreases the level of infection with some stem rust races, but races from Eastern Europe, Asia, Egypt, Ethiopia, and Turkey are virulent to it [29,30]. The source of the gene *Sr30* on chromosome 5DL is Canadian cv. ‘Webster’, which could inherit it from the Russian gene pool [60,61]. The gene is considered to confer a high level of resistance (complete immunity in case of cv. ‘Webster’) against stem rust races that are common in Europe and North America, but some Australian races are virulent to this gene [29,30]. In addition, virulence to *Sr30* was detected in races from Spain, Ethiopia, Turkey, Pakistan, and South America [30], namely, TTRTF [11], and Ug99 [15]. The *Sr41* gene on chromosome 4D of cv ‘Waldron’ has not been widely employed in breeding programs [62,63]. The gene confers juvenile and adult resistance but not against Ug99 and other races of stem rust prevalent in the world [65].

The *Sr42* gene was derived from cv. ‘Norin 40′ and mapped on chromosome 6DS [64]. At the same locus, the genes *SrCad* from cv. ‘Cadillac’ and *SrTmp* from cv. ‘Triumph 64′ were localized [41]. All three genes proved to confer resistance against the race TTKSK of the stem rust group Ug99 but among them only *SrCad* confers resistance against other deleterious races, as TTRTF and some others are virulent to *SrTmp* [11,20,41]. On the other hand, juvenile resistance conferred by *SrCad* is expressed on a sufficient level only in plants with the resistance allele of the *Lr34/Yr18/Pm38/Bdv1/Sr57* gene [41,70]. Moreover, the *SrCad* gene is associated with the *Bt10* gene conferring resistance to common bunt caused by *Tilletia tritici* (Bjerk.) G. Winter [41,69]. Among the genes, only for *SrCad* molecular markers for the resistance allele were developed [69,71].

The *Sr48* gene on chromosome 2AL originated from cv. ‘Arina’ [66]. It was considered to confer moderate but stable juvenile resistance against Ug99 races as well as other stem rust races [17]. It was revealed that the gene is quite common among Australian wheat cultivars [66]. Although there are no open sources with molecular markers linked to it, the linkage of the gene with the yellow rust resistance gene *Yr1* and microsatellite markers *sun590* and *sun592*, being the closest ones, was reported [66,74].

The *Sr49* gene was detected in cv. ‘Mahmaudi’ from Tanzania [66]. It confers resistance against all Australian stem rust races but not against Ug99 [17]. This gene is effective against the race TTRTF but new Spanish races with virulence to this gene have been recently found [12]. The *Sr54* gene was localized on chromosome 2DL of cv. ‘Norin 40′ but was not studied due to its low effectiveness against Ug99 and other modern races of stem rust [68].

APR genes should be mentioned separately as they confer a moderate but stable level of resistance against one or several pathogens with low or moderate infection loads and can increase manifestation of other resistance genes [75] (Table 2). Another benefit of APR genes is their effectiveness over a long period of time and the fact that there are no races of the pathogens that completely overcome them [76]. One of the most studied is the *Lr34/Yr18/Pm38/Bdv1/Sr57* gene on chromosome 7DS, which confers moderate resistance to all rust species, powdery mildew, and barley yellow dwarf virus [76]. In addition, the *Lr34/Yr18/Pm38/Bdv1/Sr57* gene was shown to enhance expression of other known and unknown factors of resistance against stem rust [70], in particular, Ug99 [15,41]. The gene was sequenced and shown to code for a pleiotropic drug resistance-like (PDR-like) ATP-binding cassette (ABC) transporter involved in abscisic acid signaling [77,78]. Codominant and dominant markers *cssfr5*, *SNP12*, and *ISBP1* for the resistance-associated allele have been proposed [77,79].

Another APR factor, the *Lr67/Yr46/Sr55/Pm46/Ltn3* gene, is located on chromosome 4DL [80]. The gene was first identified in the common wheat line PI250413, and the line based on cv. ‘Thatcher’ with the gene was developed [41,80]. The gene was shown to confer moderate resistance against the stem rust races of the Ug99 group [81]. The sequencing of the *Lr67/Yr46/Sr55/Pm46/Ltn3* gene revealed that it encodes a hexose transporter [86]. The *Sr56* gene was discovered in cv. ‘Arina’ and localized on chromosome 5BL [82]. It confers APR that decreases stem rust infection by 12–15% [83]. Another APR gene, *Lr46/Yr29/Pm39/Sr58*, was localized on chromosome 1BL of cvs. ‘Pavon 76′ and ‘Lalbahadur’ [85,87].

## 3. Introgressed Stem Rust Resistance Genes

The diversity of *Sr* genes in the bread wheat gene pool was substantially enriched by the transfer of *Sr* genes from species belonging to its primary, secondary, as well as tertiary gene pools [88] (Table 3). The *Sr11* gene was introgressed from cv. ‘Gaza’ of the tetraploid wheat *T. turgidum* ssp. *durum* [32]. The gene was localized on chromosome 6BL [62]. Stem rust races that are virulent to this gene may be encountered in Australia [89], South Africa [90], Canada [91], and the USA [92] but the gene is considered to confer resistance against the races common in Europe and India [29,30]. There were also reports about a low level of virulence to this gene in the races of *P. graminis* common in China and some regions of Africa [90].

The *Sr12* gene was transferred to cv. ‘Marquillo’ and afterwards ‘Thatcher’ from cv. ‘Iumillo’ of *T. turgidum* ssp. *durum* [95]. The gene had been sufficiently effective until 1950 when the especially virulent stem rust race emerged [5]. However, there is evidence that the resistance level of *Sr12* carriers significantly increases in the presence of other resistance genes, such as *Sr9* [92].

The *Sr13* gene was introgressed into common wheat cv ‘Khapstein’ from *T. turgidum* ssp. *dicoccum* cv ‘Khapli C.I.4013’ [95] to chromosome 6AL [36]. The gene is temperature-sensitive (the highest resistance level was observed at 20–28 °C) and confers moderate resistance against stem rust races that are common in Pakistan and India, but the races found in Europe and North America are highly virulent to this gene [29,30]. The *Sr13* gene is considered to confer juvenile resistance against all the races of the Ug99 group [96]. A precise and convenient molecular marker for *Sr13* was developed [132]. The cloned candidate gene encodes the CNL13 protein containing coiled-coil (CC), nucleotide-binding (NB), and leucine-rich repeat (LRR) domains, which are traditional for juvenile race-specific gene products. [96]. However, the authors indicated that the *Sr13*-mediated resistance is associated with the slower growth of the pathogen at high temperatures but not with the rapid cell death typical for the hypersensitivity response. In the later research, four different haplotypes of the gene, R1 (allele *Sr13a*), R2 (*Sr13b*), R3 (*Sr13c*), and R4 (*Sr13d,* susceptible to TTKSK and some other races tested) were discovered with use of more precise markers. The haplotypes provide different levels of resistance to some races of stem rust at different temperatures [133]. *Sr13* alleles differ in reaction to the race TTRTF, which is virulent on *Sr13b* and avirulent on *Sr13a* [23].

The *Sr14* gene was transferred from *T. turgidum* var. *dicoccum* cv. ‘Khapli’ to hexaploid cv. ‘Steinwedel’, and as a result cv. ‘Khapstein’ was obtained [44]. The gene was localized in the pericentromeric region of chromosome 1BL [134,135]. The resistance associated with *Sr14* manifests at high temperatures and with good lighting [29,44]. The gene does not confer resistance to the races common in the USA [30]. The gene *Sr17* on chromosome 7BL is one of the few recessive resistance genes and the only such gene that confers resistance to stem rust [136]. The gene originates from *T. turgidum* ssp. *dicoccum* cv. Yaroslav. It was introgressed into cv. ‘Hope’ and line ‘H-44′ together with other resistance genes [93]. The gene is effective at temperatures lower than 25 °C [136]. There are stem rust races worldwide that are virulent to this gene [29,30,73].

The *Sr21* gene can be found in the samples of *T. monococcum* L. including line ‘Einkorn C.I.2433′, which is used as a differentiator [97,137]. This gene is more effective at higher temperatures (20–24 °C) than at lower temperatures (16 °C) [138]. *Sr21* was localized on chromosome 2AL [138]. The races virulent to *Sr21* are often met in North [73] and South America [30], there are also virulent mutants in Australia [74]. However, the gene confers resistance to most races of Ug99 [75,138]. To identify *Sr21*, the molecular markers *FD527726* (0.15 cM distal), *EX594406* (0.05 cM, proximal), and microsatellite *Xgwm312* can be used [138]. The gene encodes a CC-NB-LRR protein (NLR) and accounts for the upregulation of multiple pathogenesis related genes at high temperatures [139].

The *Sr22* gene was derived from *T. monococcum* ssp. *boeoticum* and localized on chromosome arm 7AL of bread wheat [140]. The gene is temperature-sensitive and effective at lower temperatures [5], and the *Sr22b* allele was recently identified [141]. *Sr22* is effective against the Ug99 races but is linked with genes affecting agronomic traits of wheat [75,142]. The markers *Xcfa2123*, *Xwmc633*, *XcsIH81*-*BM*, and *XcsIH81-AG* were identified as closely linked to the gene, the latter is considered to be a diagnostic marker [142]. *Sr22* was cloned and its product was shown to contain typical CC, NB, and LRR regions although the gene was shown to belong to a small gene family with three homologs [143]. Fourteen orthologs of the *Sr22* gene were later reported both in *T. boeoticum* and other related species including *T. aestivum* although only one of them seemed to confer resistance to *P. graminis* [144].

The *Sr24* gene was transferred to chromosome 3DL of bread wheat from the wild grass *Thinopyrum ponticum* (Podp.) Z.-W.Liu and R.-C.Wang (syn. *Agropyron elongatum* (Host) P. Beauv.) [100,145]. Although stem rust races with virulence to this gene were detected in South Africa [90], India [146], and Australia [5], the gene confers resistance against the majority of widespread *P. graminis* races [29,30], including the race TTRTF [11]. Moreover, *Sr24* confers resistance against some of the Ug99 races [15,21]. The marker *Sr24#12* is suggested to be completely linked to the gene [147]. Another gene conferring resistance against Ug99, is *Sr25* cosegregating with *Lr19* on the chromosomal fragment introgressed from *Th. ponticum* to chromosome 7DL [101,102]. The gene is not widely involved in breeding programs because it is linked with an undesired phenotypic trait, the yellow flour color [102,148]. The marker *BF145935* was validated for the *Sr25* gene [149]. *Sr26* was also transferred from *Th. ponticum* [103]. The translocation is suggested to be full (6A/6Ag) but there is also evidence for a partial exchange with arms 6AL or 6AS [150]. The *Sr26* gene confers resistance against the Ug99 races [15]. The translocation of the 6AS/6AL-6Ae#1L segment with the gene may cause a 9% decrease in yield [99]. Nevertheless, this gene is employed for improving stem rust resistance [151]. A codominant system to detect the gene was developed. The marker *Xsr26#43* produces a 233-bp fragment in case of resistance, and marker *XBE51879* produces a fragment of 328 bp if the gene is absent [152]. The *Sr26* gene was cloned and found out to belong to the NLR family such as the majority of cloned wheat resistance genes [127].

The *Sr27* gene originated from rye (*S. cereale*) cv. ‘Imperial’ and was localized on the 3A-3R translocation [104]. It is effective against Ug99 races [15,54] and virulence to this gene is quite rare as it can be found among greenhouse mutants of *P. graminis*, hybrids created in a laboratory, as well as African races including the ones from the Ug99 group [153]. Since the mid-1950s, the rye 1RS arm carrying the stem rust resistance gene *Sr31* has been introgressed into many wheat cultivars [105,154]. The gene conferred resistance to all known races of stem rust at the time [155]. The SCAR markers for the translocation with the gene were developed [154]. However, TTKSK and other *P. graminis* races of the Ug99 group are able to infect plants with *Sr31* [15]. Nevertheless, this gene proved to be effective against the race TTRTF. In 2022, a novel Spanish stem rust race TKHBK with virulence to *Sr31* was reported [26]. Moreover, Patpour et al. [12] described 22 Spanish races with virulence to *Sr31,* of which 14 were found on bread wheat and the others on rye and *Elymus repens,* in areas of proximity to barberry, and one race with virulence to *Sr31* (LTBSK) in Western Siberia.

The *Sr32* gene was introgressed from *Aegilops speltoides* Tausch independently in several cases into all the group 2 chromosomes [106,156]. According to previous research in the USA, Canada, Mexico, and Southern Africa, no races virulent to the gene were discovered [136] until recently when a race with virulence to *Sr32* was detected in Kazakhstan [7]. The gene was not widely used in breeding programs because of the trait adherent glume and other harmful traits [156]. The markers *Xstm773* and *Xbarc55* are considered to be diagnostic markers for this gene [106].

The *Sr33* gene was first localized in *Ae. tauschii* Coss. and then transferred to chromosome 1DS of bread wheat [107,157]. The *Sr33* gene is effective against Ug99 races of stem rust [15]. No race virulent to the *Sr33* gene had been discovered during the field trials of wheat samples with the gene [30,54] until recently in Spain [26]. The gene was discovered to confer a significantly higher level of resistance when expressing in diploid plants (*Ae. tauschii*), while in *T. aestivum* the level of race-specific resistance was evaluated as moderate resistance—moderate susceptibility in the case of especially harmful races [54]. The study of expression of the gene revealed that it manifests itself as a common race-specific R-gene conferring resistance to biotrophic pathogens forcing hypersensitive cell death to prevent disease spread and feeding [158]. The first molecular markers that were considered to be effective for detection of the *Sr33* gene were *Xbarc152* and *Xcfd15* [159]. The *Sr33* gene and other closely linked genes *AetRGA1a-d*, *AetRGA2a*, and *AetRGA3a* were further cloned; the gene was found to contain six exons and the protein had the structure common for factors of juvenile resistance including a nucleotide binding site, an N-terminal CC, and a C-terminal LRR [160]. In further studies of the CC domain of SR33, additional similarity of its spatial structure to other proteins associated with race-specific resistance was discovered, and hypotheses of hypersensitive response triggered by this, and other proteins were confirmed [161]. Furthermore, the SR33 protein is homologous to MLA34 of barley conferring resistance against powdery mildew (*Blumeria graminis* f. sp. *hordei*) and TmMLA, an MLA-like protein of *T. monococcum* (the homology level in both cases is up to 86%), while the homology level for earlier discovered resistance associated proteins of wheat LR1, LR10 and LR21 was fairly low [160]. The SR33 protein, unlike proteins analogues, does not need chaperone proteins [161].

The *Sr34* gene was transferred from the wild relative *Ae. comosa* Sibth. and Sm. to chromosome 2A (the 2A-2M translocation) and 2D (the 2D-2M translocation) together with the yellow rust resistance gene *Yr8* [93]. *Sr34* is considered to be more effective at lower temperatures. Avirulence to this gene is not common in Australia but more intrinsic for stem rust races from Southern Asia, China, Ethiopia, Kenya, and South America [29,30,152]. However, the gene does not confer resistance against Ug99 races [54].

The resistance gene *Sr35* was transferred from *T. monococcum* to chromosome 3AL of common wheat [162] and confers resistance against the race TTKSK (Ug99) of stem rust and its variants TTKST and TTTSK [15,21,54]. The gene proved to confer resistance to moderate resistance under the infection background with a comparatively mild course of the disease during the field trials in Kenya in 2005–2006 [54]. *Sr35* confers resistance to *P. graminis* races common in Australia and North America but there are races virulent to the gene in Ethiopia, Kenya, Malaysia, Nepal, Brazil, Chile, Argentina, and China [30]. The races TTRTF and TTKTF of clade IV-E2, which are currently found in Europe, are also virulent to this gene [12,23]. The markers *XAK335187* and *Xcfa2170* are considered to be closely linked with this gene [163]. *Sr35* was sequenced and identified as identical to the *CNL9* gene candidate having a 196 bp 5′UTR and a 1526 bp 3′UTR that includes three introns. Orthologs for the gene as well as for *Sr13* were identified based of DNA sequences from related species [163].

The *Sr36* gene was introgressed from *T. timopheevi* on chromosome 2BS and first designated as the allele *Sr9c* [109]. It confers resistance against the original Ug99 race of *P. graminis* but virulence of other races from the Ug99 group was reported [15,16,17,18,19,21]. Furthermore, cases of serious infestation with stem rust of cultivars with this gene in Australia and North America were reported, and other isolates virulent to *Sr36* were also revealed [30]. *Sr37* is another gene that is transferred from *T. timopheevi* [110]. The translocation with the gene and other potential resistance factors was localized on chromosome 4B but it has not gained major distribution [5]. According to the literature, races of stem rust virulent to the gene are quite common [29,30] and include TTRTF [23]. However, this gene is effective against Ug99 races [164].

The *Sr38* gene was localized on a 2AS-2NS translocation from *Ae. ventricosa* Tausch together with the leaf rust resistance gene *Lr37* and the yellow rust resistance gene *Yr17* [111]. Virulence to *Sr38* was first detected in 2000–2001 in South Africa and since then, in addition to the Ug99 group, it has been defeated by many stem rust races in European and Asian regions [12,165].

The *Sr39* gene was localized on arm 2S introgressed into chromosome 2B of wheat from *Ae. speltoides* [112]. On the same arm at a distance of 3 cM, the leaf rust resistance gene *Lr35* was localized [166]. The *Sr39* gene confers juvenile and adult plant resistance at a level of resistance to moderate resistance to all races of stem rust known at the moment, including Ug99 [15,54]. RL6082 and other wheat lines obtained as a result of introgression showed a significant increase in flour water absorption and significant degradation of flour quality and other agronomic traits [112,163]. Therefore, attempts to decrease the size of the introgressed fragment keeping the *Sr39* and *Lr35* genes were made [167,168]. In particular, based on line RL5711 with the 2B-2S translocation, lines #220 and #247 were developed with the gene and the markers *Sr39#50s* and *Sr39#22r* for the *Sr39* gene were developed [167]. Line #247 was used in breeding programs in Australia [167]. Other authors took RL6082 as a basis and with the use of the improved scheme for chromosome engineering obtained red wheat lines RWG1, RWG2, and RWG3 with reportedly small parts of the chromosome 2S. The authors proposed the markers *Xrwgs27*, *Xrwgs28*, and *Xrwgs29* as more convenient for breeding [167]. Further validation of the markers for the *Sr39* gene was carried out with use of a large amount of cultivars and lines. Based on this, *Sr39#50s* and *Sr39#22r* were considered precise and convenient but *Xrwgs27*, *Xrwgs28* and *Xrwgs29* were not precise enough or produced amplified fragments that were hard to distinguish with electrophoresis. Addtionally, the marker *Xwmc474* was described and considered to be a diagnostic marker for the *Sr39* gene [169].

The *Sr40* gene was introduced from *T. timopheevii* subsp. *armeniacum* (Jakubz.) Slageren on translocation 2B-2G#2S [113]. The resulting line RL6088 was considered to be resistant against all the races of stem rust, including Ug99 [15,54]. However, a unique race with virulence on *Sr32* and *Sr40* was recently reported in Kazakhstan [7]. The line RL6088 was used to map the introgressed arm and the markers *Xgwm319*, *Xwmc344*, *Xwmc474*, *Xwmc477*, *Xgwm374*, and *Xwmc661* were discovered to be linked to the gene of interest [114]. None of the markers were validated by other authors although it was claimed that the marker *Xsr39#22r* could detect the *Sr40* gene in the breeding material that does not carry the *Sr39* gene [169]. The fragment of the chromosome with the *Sr40* gene also carries traits affecting flour quality and other agronomic traits [114].

The *Sr43* gene was first discovered in *Th. ponticum* due to its effectiveness against stem rust and then successfully transferred to common wheat on the translocation 7DS-7el2L [115,170]. The gene has not gained major distribution because the translocated chromosome arm also carried factors affecting agricultural quality, in particular for yellow flour color [115]. The *Sr44* gene derives from a partial amphyploid of wheat with the wild relative *Thinopyrum intermedium* (Host) Barkworth and D.R. Dewey (translocation 7DL-7J#1S) and confers resistance against all the biotypes of Ug99 [116]. However, Australian and European races of stem rust with virulence to this gene were discovered [12,116].

The *Sr45* gene on chromosome 1DS was introgressed from *Ae. tauschii* [117,171]. It confers resistance against all the races of Ug99 and stem rust races that are common in India and Australia, but virulence to this gene was reported for races in Canada [117] and the novel Spanish race TKGLK [12]. The gene was cloned with use of the same technique as *Sr22* and was also found to encode a CC-NB-LRR protein belonging to a family with 8–12 homologs [143]. The *Sr46* gene was transferred from *Ae. tauschii* var. *meyeri* on chromosome 2DS. Virulence to this gene was discovered among stem rust races distributed over the world. Tests with Ug99 and wheat lines with *Sr46* have been not carried out although the samples of *Ae. tauschii* with the gene showed juvenile resistance against the race TTKSK [118]. The gene was sequenced by conventional fine mapping in segregating diploid progenitor and wheat populations coupled with the sequencing of candidate genes in this region and was discovered to encode a CC-NB-LRR protein [172]. The *Sr47* gene was introgressed from *Ae. speltoides* on the translocation 2BL-2SL 2SS [119,120]. During all the field trials it conferred total resistance against stem rust, but trials with Ug99 have not been carried out [120].

The *Sr50* gene originates from rye cv. ‘Imperial’ (translocation 1DL.1DS-1R#3S-1DS) and confers resistance to Ug99 races but is sensitive to some other common races of *P. graminis* [121], in particular to TTRTF and the novel Spanish race KKGBM [12]. The gene was cloned and found out to be homologous to the barley *Mla*, encoding a CC-NB-LRR protein. The resistance conferred by it was discovered to be different from *Sr31* and other genes on rye chromosome 1RS and molecular genetic markers for the gene were described [173]. The gene product was shown to interact with the pathogen in a way that has been described for CC-NB-LRR proteins and involved recognition of the corresponding *AvrSr50* product. *P. graminis* races virulent to the gene were shown to express the protein with a substitution in a single surface-exposed residue or did not express it at all due to mutations in the gene [174].

The *Sr^Amigo^* (*Sr1RS^Amigo^*) gene was introgressed as part of the 1AL/1RS translocation from the Argentinian rye cv. ‘Insave’ [131]. The first cultivar with the translocation was ‘Amigo’ registered in the USA in 1976, which obtained the translocation from the octoploid triticale cv. ‘Gaucho’ [175]. The presence of the translocation with the gene can be easily detected by electrophoresis of storage proteins. The *Sr^Amigo^* gene confers moderate race-specific resistance against biotypes of Ug99 but is not effective against some other races of stem rust, in particular TRTTF [176] and TKKTP [9,173]. However, according to Patpour et al. [12], TKKTP is avirulent to this gene.

The *Sr51* gene was transferred from *Ae. searsii* Feldman and Kislev ex K. Hammer on translocations 3AL-3S^S^S, 3BL-3S^S^S, and 3DL-3S^S^S and translocation recombination 3DS-3S^S^S/3S^S^L [122]. The gene confers total resistance against Ug99 and other races of stem rust it was trialed with, but the work to localize and introduce it on a smaller chromosome fragment is still in progress [122]. 

The *Sr52* gene originates from the wild grass *Dasypyrum villosum* (L.) Borbas (translocation 6AL-6V.3L) [123]. It confers temperature-sensitive (effective within the temperature range 18–26 °C) resistance against the original race of Ug99 and other races of stem rust. In addition, possible coexpression with the increased level of resistance between this gene and others on the translocated arm or those of *T. aestivum* was reported [123]. However, it is ineffective against many currently prevalent European stem rust races [12].

The *Sr53* gene was obtained from *Ae. geniculata* Roth by translocation of the chromosome 5M(g)L/5M(g)S part to arm 5DL [124]. The gene confers moderate juvenile as well as adult resistance against all the races of stem rust, including Ug99. The *Sr59* gene was transferred from the rye *S. cereale* on the 2DS·2RL Robertsonian translocation, mapped, and proved to confer resistance against the Ug99 races of stem rust [125]. 

The *Sr60* gene was transferred to common wheat from *T. monococcum* chromosome 5A^m^S. It is effective against the races QFCSC, QTHJC, and SCCSC but not the Ug99 group. The gene is closely linked to the markers *CJ942731* and *GH724575* and completely linked to *LRRK123.1* [126]. *Sr60* is 5008 bp in length with a complete coding sequence of 2175 bp; the predicted protein is 724 amino acids long and unlike most of the sequenced wheat resistance genes contains two putative protein kinase domains [126]. The *Sr61* gene had been previously designated as *SrB* and was, such as *Sr26*, transferred from *Th. ponticum*. The gene proved to confer resistance to a set of races of stem rust other than *Sr26* [127]. Similar to *Sr26,* it was also sequenced and found to be of the NLR type [127]. The *Sr62* gene was introgressed into common wheat from *Ae. sharonensis* Eig on translocation 1S^sh^S·1S^sh^L-1BL/1S^sh^S·1S^sh^L-1DL [128,129]. The gene proved to confer resistance against some races of the Ug99 group. It was mapped, and KASP markers were developed [177].

Resistance conferred by the *Sr63* gene was first described in *T. turgidum* cv. ‘Glossy Huguenot’ as an APR [178]. Recently the gene was mapped to chromosome 2AL, proved to confer resistance against all races of stem rust including Ug99, and closely linked molecular markers for its detection were developed [130]. Another gene conferring race nonspecific APR against stem rust and race-specific juvenile resistance against leaf rust as well as tolerance against powdery mildew is *Sr2/Lr27/Pbc* [93,94]. The gene was transferred from emmer wheat (*T. turgidum* ssp. *dicoccum*) cv. ‘Yaroslav’ in the 1920s and was reported by McFadden in 1930. As a result, cv. ‘Hope’ was obtained [93]. Further, it was discovered that APR in cv. ‘Hope’ was conferred by a single gene designated as *Sr2* [179]. Resistance conferred by the *Sr2* gene is associated with a decreased number of uredinia in infected plants. The gene was discovered to provide the highest level of resistance at the flowering stage [180]. The gene remains effective for more than 80 years and no stem rust race, including Ug99, is virulent against it [94,181]. In addition, the ability to enhance unknown factors of stem rust resistance was discovered for this gene, as with other APR genes [182]. Identification of resistance conferred by the gene in the field is complicated due to the moderate level of expression [183,184]. The main morphological trait of *Sr2* is pseudo black chaff, which was reported to manifest itself unevenly in the field depending on other genes and temperature [184]. The juvenile leaf rust resistance gene *Lr27*, which was discovered to cosegregate with *Sr2*, needs resistance associated with the *Lr31* gene to fully manifest itself [94].

The *Sr2* gene was localized on the short arm of chromosome 3B [185]. On the first genetic map, the *Lr27* and *Sr2* genes were mapped at some genetic distance [183]. *Gwm533* was the first molecular marker with a sufficient level of polymorphism, which was used in breeding for the resistance associated allele. It has three alleles: the 0 and 155 bp alleles are mostly associated with a lack of resistance and the 120 bp allele is mostly associated with resistance conferred by the gene [185]. Further, based on the marker sequence, STS markers *stm598ctac* and *stm598gtag* were developed; the markers have several alleles and only one for each marker (56~61 bp and 83~85 bp, respectively) was associated with resistance [186]. Additionally, based on the BAC library of wheat, a more detailed genetic map for chromosome 3B, including the *Sr2* region, was developed and subsequently, more tightly linked markers were discovered: *BE426676*, *BE401794*, *BE500189*, *CA640157*, and *BE591959*. Among them, *BE500189* and *CA640157* showed the closest linkage, 0.14 cM and 0.07 cM, respectively [187]. Moreover, a number of SSR markers *3B028F08*, *3B042G11*, and *3B061C22* were developed based on the BAC map [188]. The SSR markers are close enough to the *Sr2* gene, but they were found to have different alleles for the samples with the same allele of the *Sr2* gene during trials with a number of lines and cultivars [169]. Recently, the SCAR marker *csSr2* was identified based on the data from the BAC library of cv. ‘Hope’ 3B chromosome partial sequence. It was discovered that resistance and susceptibility-associated alleles differ by a single nucleotide polymorphism and in case of the resistance-associated allele there is a restriction site for enzyme BspHI; the marker is precise and does not segregate with the *Sr2* gene (the estimated degree of accuracy is around 95%) [94]. More detailed mapping of the *Sr2/Lr27/Pbc* locus was carried out and the candidate genes that could confer *Sr2-*like resistance were proposed namely: *TaGLP3_1/TaGLP3_2*, *TaGLP3_3/TaGLP3_4*, *TaGLP3_10*, *TaGLP3_6*, *TaGLP3_5*, *TaGLP3_8/TaGLP3_9*. The location of the marker *csSr2* was further revised and another marker was identified [118].

## 4. Conclusions

Numerous genes conferring resistance to stem rust have been discovered to date either in the gene pool of common wheat or in related species. Although many of them do not confer resistance to Ug99 races, some, such as *Sr31*, are still effective against other numerous modern and devastating races of *P. graminis* [11,23]. On the other hand, other genes that were considered to not be effective against modern races of stem rust appeared to have some value conferring resistance against Ug99 [56]. In accordance with recent discoveries, the Ug99 races of *P. graminis* should not be the only races considered as a threat to wheat production worldwide [3,9,12,25,26]. Introgressed R-genes, such as *Sr33*, seem to be the most effective at providing “strong” types of resistance against almost all the races of stem rust known to date [160] Such resistance is based on a gene-for-gene interaction with the pathogen and recognition of the pathogen’s effectors. However, as effectors are usually encoded by a single gene, mutations often cause an emergence of new races virulent to race-specific genes [1,6,7,174]. Resistance conferred by *Sr2,* as well as other APR genes, is based on more elaborate mechanisms of physiological responses to the pathogen invasion in general, therefore it seems to be far more promising on a long-term scale [76,189].

The studies of QTLs related to stem rust resistance in modern wheat cultivars mostly revealed the effectiveness and coexpression of known genes against the races used, but potential loci related to *P. graminis* resistance were discovered as well [181,190,191]. For instance, stem rust resistance of the spring wheat line ‘CI 14275′ to the races TTKSK, TRTTF, TPMKC, TTTTF, and RTQQC was shown to be conferred not only by the *Sr12* gene but also by the unknown QTLs *QSr.cdl-2BS.2* and *QSr.cdl-6A* [192]. Based on RIL populations from cv. ‘Baguette 13′, cv. ‘INIA Tero’, and line BR23//CEP19/PF85490′, QTLs on chromosomes 2B, 6A, and 7B were identified, of which the QTL on 2B was effective against Ug99 [59]. The *QSr.umn-2B.2* QTL on chromosome 2B conferring APR against African and North American stem rust races (including the Ug99 race group) in four environments in the RB07/MN06113-8 population has been reported in the literature [166]. In addition, several minor QTLs on different chromosomes and major QTLs on chromosomes 1A and 1B were discovered [193].

To date stem rust resistance genes such as *Sr13*, *Sr21*, *Sr22*, *Sr26*, *Sr33*, *Sr35*, *Sr45*, *Sr46*, *Sr50*, *Sr55*, *Sr57*, *Sr60*, and *Sr61* were sequenced [77,86,96,126,127,143,163,172,173,174]. Of them, only two (*Sr55* and *Sr57*) are APRs and bread wheat own genes. Considering the importance of not only using stem rust resistance genes in breeding but also understanding the mechanisms of resistance, such studies are of prime value. New approaches such as “rapid cloning”, which was used for the *Sr22* and *Sr45* genes [143], and “sequence capture” which was used to clone *Sr46* [172], might be considered as a way forward to sequence other stem rust resistance genes for studying special features of the expression of *Sr* genes in response to the pathogen and their involvement in plant immunity mechanisms.

Stem rust caused by *P. graminis* remains a constant threat to agriculture worldwide. New races have emerged that are virulent to the resistance genes which were considered to be effective even several decades ago. On the other hand, some *Sr* genes previously considered as ineffective have proved to be of use as they might provide resistance to new exotic races of stem rust. Moreover, when pyramided with other race-specific genes or genes conferring race-nonspecific moderate APR any resistance gene may take part in complex and durable stem rust resistance. The search for new *Sr* genes, especially APR genes, continues to be of primary importance. In this context, the report about the identification of the *SrPan3161* gene on chromosome 4D in bread wheat cultivar PAN 3161, which accounts for 71.5% of the phenotypic variation for field resistance to the Ug99-group race PTKST, is very promising. This gene derives from the cultivar Tugela and may represent a novel APR [194].

## Figures and Tables

**Table 1 pathogens-11-01157-t001:** Own race-specific stem rust resistance genes in bread wheat.

Gene	Allele	Resistance Against Ug99	Possible Source	Chromosome	DNA Marker Available	References
*Sr5*	-	No	Kanred	6DS	No	[5,29,30]
*Sr6*	-	No	McMurachy	2D	No	[5,31]
*Sr7*	*a*	No	Ciano-67	4AL	No	[32,33,34,35]
	*b*	No	Selkirk			
*Sr8*	*a*	No	Frontana	6AS	No	[36,37]
	*b*	No	Bezostaya 1			
*Sr9*	*a*	No	TAM-107	2BL	Yes	[38,39]
	*b*	No	Chinese spring			
	*e*	No	Arrivato			
	*f*	No	Chinese white			
	*h*	Yes	Webster			[40,41]
*Sr10*	*-*	No	Marquiz	2B	No	[40,42]
*Sr15*	*-*	Maybe	Norka	7AL	Yes	[43,44,45,46]
*Sr16*	*-*	No	Reliance	2BL	No	[38,47]
*Sr18*	*-*	No	Gabo	1DL	No	[48,49]
*Sr19*	*-*	No	Marquiz	2B	No	[50]
*Sr20*	*-*					
*Sr23*	*-*	No	Myronovskaya 264	2BS	Yes	[51,52]
*Sr28*	*-*	Moderate	Kota	2BL	Yes	[53,54,55,56,57]
*Sr29*	*-*	No	Aurora	6D	No	[58,59]
*Sr30*	*-*	No	Webster	5DL	No	[60,61]
*Sr41*	*-*	No	Waldron	4D	Yes	[62,63]
*Sr42*	*-*	No	Norin 40	6DS	Yes	[64,65]
*Sr48*	-	Moderate	Arina	2AL	?	[66]
*Sr49*	-	No	Mahmoudi	5BL	?	[67]
*Sr54*		No	Norin 40	2DL	?	[68]
*SrCad*	*-*	Yes	Cadillac	6DS	Yes	[41,69,70,71]
*SrTmp*	*-*	Yes	Triumph 64		Yes	[41,71]

**Table 2 pathogens-11-01157-t002:** Own race-nonspecific stem rust APR genes of common wheat.

Gene	Cosegregating Resistance Factors	Resistance Against Ug99	Possible Source	Chromosome	DNA Marker Available	References
*Sr55*	*Lr67/Yr46Pm46*	Yes	PI250413	4DL	Yes	[41,80,81]
*Sr56*	-	Yes	Arina	5BL	Yes	[82,83]
*Sr57*	*Lr34/Yr18/Pm38/Bdv1*	Yes	Bezostaya 1	7DS	Yes	[77,79,84]
*Sr58*	*Lr46/Yr29/Pm39*	Yes	Pavon 76	2D	Yes	[85]

**Table 3 pathogens-11-01157-t003:** Introgressed stem rust resistance genes of bread wheat.

Gene	Cosegregating Resistance Genes, or Genes on the Same Arm	Resistance Against Ug99	Source Species	Possible Source Cultivar	Chromosome	DNA Marker Available	References
*Sr2*	*Lr27*	Moderate	*T. turgidum* ssp. *dicoccum*	Hope	3BS	Yes	[93,94]
*Sr9d*	-	No	*T. turgidum* ssp. *dicoccum*	NIL-LMPG-*Sr9d*-TR.DR	2BL	Yes	[93]
*Sr9g*	-	No	*T. turgidum* ssp. *durum*	-	2BL	Yes	[39]
*Sr11*	-	No	*T. turgidum* ssp. *durum*	Gaza	5BL	Yes	[32,62]
*Sr12*	-	Moderate	*T. turgidum* ssp. *durum*	Marquillo	3BS	No	[92,95]
*Sr13*	-	Yes	*T. turgidum* ssp. *dicoccum*	NIL-Marquis-*Sr13,Sr14*-Khapstein	6AL	Yes	[36,96]
*Sr14*	-	No	*T. turgidum* ssp. *dicoccum*	NIL-Marquis-*Sr13,Sr14*-Khapstein	1BL	Yes	[44]
*sr17*	-	No	*T. turgidum* ssp. *dicoccum*	Selkirk	7BL	No	[93]
*Sr21*	-	Yes	*T. monococcum*	Einkorn C.I.2433	2AL	Yes	[97,98]
*Sr22*	-	Yes	*T. monococcum* ssp. *boeoticum*	Schomburgk	7AL	Yes	[99]
*Sr24*	-	Yes	*Thinopyrum ponticum* (Podp.) Z.-W.Liu and R.-C.Wang	NIL-LMPG-*Sr24*	3DL	Yes	[100]
*Sr25*	*Lr19*	Yes	*Ag. elongatum* Host. *(Th. ponticum)*	NIL-LMPG-*Sr25*	7DL	Yes	[101,102]
*Sr26*	-	Yes	*Ag. elongatum* Host. *(Th. ponticum)*	NIL-LMPG-*Sr26*	6A/6Ag	Yes	[103]
*Sr27*	-	Yes	*S. cereale*	NIL-LMPG-*Sr27*	3A/3R	No	[104]
*Sr31*	*Lr26*, *Yr9*	No	*S. cereale*	Knyahynia Olha	1BL/1RS	Yes	[105]
*Sr32*	-	Yes	*Aegilops speltoides* Tausch	-	2A, 2B, 2D	Yes	[106]
*Sr33*	*Lr21*	Yes	*Ae. tauschii* Coss.	Lorikeet	1DS	Yes	[107]
*Sr34*	*Yr8*	No	*Ae. comosa* Sibth. and Sm.	Marquillo	2A/2M, 2D/2M	Yes	[108]
*Sr35*	-	Yes	*T. monococcum*	NIL-STEWART-Sr35-G-2919	3AL	Yes	
*Sr36*	-	Yes	*T. timopheevi*	Songlen	2BS	Yes	[109]
*Sr37*	-	Yes	*T. timopheevi*	Boohai	4B	Yes	[110]
*Sr38*	*Lr37/Yr17*	No	*Ae. ventricosa* Tausch	Trident	2AS/2NS	Yes	[111]
*Sr39*	-	Yes	*Ae. speltoides* R.L.5344.	RL-6082	2B	Yes	[112]
*Sr40*	-	Yes	*T. timopheevii* subsp. *armeniacum* (Jakubz.)	Maris-Fundin	2BS	Yes	[113,114]
*Sr43*	-	Yes	*Ag. elongatum* Host. *(Th. ponticum)*	RWG-33	7DS/7el2L	Yes	[115]
*Sr44*	-	Yes	*Th. intermedium* (Host) Barkworth and D.R. Dewey	Payne	7DL/7J#1S	Yes	[116]
*Sr45*	-	Yes	*Ae. tauschii*	Thornbill	1DS	Yes	[117]
*Sr46*	-	?	*Ae. tauschii* var. *meyeri*	AUS-18913	2DS	Yes	[118]
*Sr47*	-	?	*Ae. speltoides*	96–90	2BL/2SL·2SS	Yes	[119,120]
*Sr50*	-	Yes	*S. cereale* cv. Imperial	-	1DL.1DS/1R#3S/1DS	Yes	[121]
*Sr51*	-	Yes	*Ae. searsii* Feldman and Kislev ex K.Hammer	TA-6555	3AL/3S^S^S, 3BL/3S^S^S 3DL/3S^S^S	Yes	[122]
*Sr52*	-	Yes	*D. villosum* (L.) Borbas	KS-12-WGGRC-57	6AL/6V.3L	Yes	[123]
*Sr53*	-	Yes	*Ae. geniculata* Roth	KS-12-WGGRC-58-T1	5M(g)L/5M(g)S / 5DL	Yes	[124]
*Sr59*	-	Yes	*S. cereale*	TA5094	T2DS·2RL	Yes	[125]
*Sr60*	-	No	*T. monococcum*		5A^m^S		[126]
*Sr61*	-	Yes	*Ag. elongatum* Host.(*Th. ponticum*)	W3757	T6AS.6AL-6Ae#1	Yes	[127]
*Sr62*	-		*Ae. sharonensis* Eig	AS_1644	1S^sh^S·1S^sh^L-1BL/1S^sh^S·1S^sh^L-1DL	Yes	[128,129]
*Sr63*		Yes	*T. turgidum* ssp. *durum*	Glossy Huguenot	2AL	Yes	[130]
*Sr^Amigo^*	-	Yes	*S. cereale* cv. Insave	Amigo	1AL/1RS	Yes	[131]

## Data Availability

Not applicable.

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
