# Peer review of "Wheat Genes Associated with Different Types of Resistance against Stem Rust (Puccinia graminis Pers.)"

_pathogens, 2022, doi:10.3390/pathogens11101157_

Round 1
Reviewer 1 Report
Editorial Office
‘Pathogens’
Thank you so much for referring me to the review manuscript entitled ‘Wheat genes associated with different types of resistance against stem rust (Puccinia graminis Pers.)’. Wheat stem rust is one of the most devastating diseases of wheat and the use of genetic resistance has been historically the most effective and environmentally sound approach to control it. This manuscript presents a review of the effectiveness of the stem rust resistance genes that have been identified and characterized. Emphasis is put on resistance to races in the Ug99 group.
I found that the biggest limitation of this manuscript is the lack of updated information about the most recent stem rust epidemics and outbreaks that happened in many wheat-growing regions around the globe and the virulences to stem rust resistance genes that have been reported. The emphasis of this work is on Ug99, but much more has happened in the last 10 years that has impacted more drastically wheat production and Sr genes than Ug99 itself. This manuscript will get more relevant if virulence to the described genes is present in the most recent races identified. Here is a list of recent publications (ordered by year) that detail epidemics and outbreaks that occurred since 2013 and that necessarily must be part of your analysis of virulence/avirulence to the described resistance genes.
Patpour et al. 2022. Wheat stem rust back in Europe: Diversity, prevalence and impact on host resistance. Frontiers in Plant Science. https://doi.org/10.3389/fpls.2022.882440
Olivera et al. 2022. Novel and unique virulences from a sexual population of Puccinia graminis f. sp. tritici in Kazakhstan. Phytopathology. https://doi.org/10.1094/PHYTO-08-21-0320-R
Nazari et al. 2021. First Report of Ug99 Race TTKTT of Wheat Stem Rust (Puccinia graminis f. sp. tritici) in Iraq. Plant Disease 105. https://doi.org/10.1094/PDIS-02-21-0404-PDN
Shamanin et al. 2020. Stem rust in Western Siberia – race composition and effective resistance genes. Vavilov J. Genet. Breed. 24:131-138.
Skolotneva et al. 2020. Virulence phenotypes of Siberian wheat stem rust population in 2017–2018. Front. Agron. 2:6. doi: 10.3389/fagro.2020.00006.
Tesfaye et al. 2019. First report of TTRTF race of wheat stem rust, Puccinia graminis f. sp. tritici in Ethiopia. Plant Disease 104. https://doi.org/10.1094/PDIS-07-19-1390-PDN
Olivera Firpo et al. 2017. Characterization of Puccinia graminis f. sp. tritici isolates derived from an unusual wheat stem rust outbreak in Germany in 2013. Plant Pathology, 66: 1258–1266.
Newcomb et al. 2016. Kenyan isolates of Puccinia graminis f. sp. tritici from 2008 to 2014: Virulence to SrTmp in the Ug99 race group and implications for breeding programs. Phytopathology 106: 729-736.
Olivera et al. 2015. Phenotypic and genotypic characterization of race TKTTF of Puccinia graminis f. sp. tritici that caused a wheat stem rust epidemic in southern Ethiopia in 2013/14. Phytopathology 105: 917-928.
I also recommend organizing the description of the genes following the list presented in your tables (in numerical order). If you what to highlight the effect of temperature on some of the genes, maybe it is a good idea to present them in a separate table.
I found that in a few cases, review papers are cited to describe results from previously conducted research. For example, Singh et al. 2015 (review paper) has been cited when presenting results from original work conducted by others. Please, always refer to the original work when you present information in your review.
Several times in your manuscript you refer to race TTTTF as the one responsible for the epidemic in Italy in 2016. This race was wrongly typed in this publication. The race that caused the epidemic in Sicily is race TTRTF. This is a race that is now widely distributed across Europe and East Africa and was first detected in Georgia in 2015. Please, see the publications that refer to this race as it has a complex virulence spectrum impacting several important stem rust resistance genes.
I have additional comments in the attached file.

Author Response
Dear reviewer
Please see the detailed response in the file attached
Kindest regards, Authors

Reviewer 2 Report
Karelov et al. summarized the wheat genes associated with different types of resistance against stem rust. This is a timely review paper including a variety of recent advances, which will provide a good addition to our knowledge of the resistance to stem rust in wheat. I only have two minor comments listed below.
1. Some genes associated with the resistance to stem rust were cloned. I would suggest that the authors can add more details in their functions and mechanisms.
2. What are the different types of resistance against stem rust and the relationship with effective or non-effective genes, and with different races of stem rust? I suggest authors can clarify them.
Author Response

(The authors gave the same response as above.)

Reviewer 3 Report
Stem rust caused great loss to wheat yield and severely restricted wheat production. However, resistance of wheat cultivars to stem rust usually breakdown due to be neutralized by new virulence in the pathogen, which is not only a major scientific issue but also a practical problem to be studied and solved urgently. It was the safest, most effective and economical method to breed stable, effective, broad-spectrum and long-lasting varieties of wheat resistance against stem rust by using the host′s own disease-resistant genes.
In the MS, Karelov et al. revealed the genes associated with different types of resistance against stem rust, and briefly introduced their origin and usage. In my opinion, it was an interesting review paper with good information and could be accepted for publication after some minor revisions.
As a review paper, I suggest that the author should make a prospect of the future application of wheat stem rust resistance genes, which also provides direction and reference for future research.
The modern biotechnology means (such as gene editing, next-generation sequencing technology, etc.) have widely used in wheat stem rust resistance gene mining and breeding research. The author should provide some references in the MS.
Author Response

(The authors gave the same response as above.)
